# The Role of Tumor-Associated Antigen HER2/neu in Tumor Development and the Different Approaches for Using It in Treatment: Many Choices and Future Directions

**DOI:** 10.3390/cancers14246173

**Published:** 2022-12-14

**Authors:** Saleh Alrhmoun, Sergey Sennikov

**Affiliations:** 1Laboratory of Molecular Immunology, Federal State Budgetary Scientific Institution Research Institute of Fundamental and Clinical Immunology, 630099 Novosibirsk, Russia; 2Faculty of Natural Sciences, Novosibirsk State University, 630090 Novosibirsk, Russia; 3Department of Immunology, V. Zelman Institute for Medicine and Psychology, Novosibirsk State University, 630090 Novosibirsk, Russia

**Keywords:** HER2, HER2-positive cancer, antibody–drug conjugates, CAR-T cells, HER2-targeted immunotherapy, HER2-targeted vaccines

## Abstract

**Simple Summary:**

The human epidermal growth factor receptor 2 (HER2) plays a central role in the pathogenesis and development of several types of cancer. For a long time, the overexpression of HER2 by tumor cells was considered a predictor of a poor prognosis, until HER2-targeted therapy was introduced, starting with trastuzumab, which revolutionized the entire field of treatment for HER2-positive cancers and changed their prognosis, with the overexpression of HER2 becoming correlated with increased survival and better outcomes. Since then, this field has undergone tremendous developments, and many new approaches have been implemented, in particular immunotherapy. The purpose of this review is to provide a general understanding of the various approaches for targeting the HER2 molecule for cancer treatment, shedding light on some of the most promising updates in this field.

**Abstract:**

The treatment of HER2-positive cancers has changed significantly over the past ten years thanks to a significant number of promising new approaches that have been added to our arsenal in the fight against cancer, including monoclonal antibodies, inhibitors of tyrosine kinase, antibody–drug conjugates, vaccination, and particularly, adoptive-T-cell therapy after its great success in hematological malignancies. Equally important is the new methodology for determining patients eligible for targeted HER2 therapy, which has doubled the number of patients who can benefit from these treatments. However, despite the initial enthusiasm, there are still several problems in this field represented by drug resistance and tumor recurrence that require the further development of new more efficient drugs. In this review, we discuss various approaches for targeting the HER2 molecule in cancer treatment, highlighting their benefits and drawbacks, along with the different mechanisms responsible for resistance to HER2-targeted therapies and how to overcome them.

## 1. Introduction

The HER2 receptor “Human Epidermal growth factor Receptor2”, also known as CD340 or p185, received its name as a result of having a substantial resemblance to the Human Epidermal Growth Factor Receptor 1 (HER1). HER2 is also referred to as HER2/Neu, because it was derived from the rat glioblastoma cell line, a specific type of neural tumor, or ErbB-2, due to its similarity to the avian erythroblastosis oncogene B (ErbB). Overall, HER2 is a tyrosine kinase receptor, that is encoded by the ERBB2 gene, which is found on the long arm of chromosome 17 (17q12) [1,2,3]. HER2 belongs to the epidermal growth factor family of tyrosine kinase receptors (EGFR/ErbB), along with 3 other receptors: HER1 (EGFR, erbB1), HER2 (HER2/neu, EGFR-2, erbB2), HER3 (EGFR-3, erbB3), and HER4 (erbB4) [4]. Receptor dimerization, as a homodimer or heterodimer, is essential for signal transduction [5].

There are a total of 11 extracellular ligands for HER1, HER3, and HER4 [6]; however, HER2 is an orphan receptor with no defined ligand, which makes it completely dependent on the dimerizing partner for ligand-mediated signaling [5,7,8]. On the other hand, HER2 exists in an active form, similar to that of the EGFR when it binds to its ligand, which explains why HER2 is the primary signaling partner for other HER receptors, by forming heterodimers with them [9,10]. Heterodimerization with HER2 is more favored than homodimerization where, presumably, two ligands would be needed to convert the two receptors into an active state and, subsequently, initiate dimerization [5]. It is also worth mentioning that, unlike the other members, HER3 lacks tyrosine kinase activity, since its catalytic domain does not include an ATP binding site and is hence catalytically inactive. As a result of that, HER3 relies on the kinase activity of its heterodimeric partners, primarily HER2, for the phosphorylation of its tyrosine residue and the activation of downstream signaling cascades [6,11,12]. All of that brings us to the conclusion that HER2 emerges as a master coordinator of a signaling network rather than as a receptor that mediates the signaling of one specific ligand [8].

## 2. Biological Function of HER2

HER receptors are widely expressed and crucial for the normal functionality of a wide range of normal tissues including the skin, breast, and placenta, not to mention the epithelial cells of the reproductive, gastrointestinal, respiratory, and urinary tracts [13]. Studies have shown that they are also essential for the development of many organ systems, such as the skin, brain, lungs, and gastrointestinal tract [11].

Activation of HER2 signaling pathways leads to the phosphorylation of tyrosine residues, which, in its turn, triggers subsequent signaling cascades such as Ras/MEK/ERK, JAK/STAT, and PI3K/AKT. This evolutionarily conserved signaling module mediates a variety of cellular processes, including cell proliferation, differentiation, motility, adhesion, migration, invasion, resistance to apoptosis, and survival [4,11,14].

Gene-targeting experiments have provided proof for the crucial biological role of HER2 for proper functioning, as mice with kinase dead mutations or erbB2-null mice have been shown to experience embryonic lethality as a result of significant cardiovascular abnormalities. Furthermore, the erbB2-null mice revealed a crucial function for HER receptors in the maintenance and development of the nervous system [9,15,16].

## 3. Relevance to Cancer

The significance of this receptor stems from the observation of its overexpression in many cancer types. It is also worth mentioning that HER2 overexpression is associated with a more aggressive disease, a greater recurrence rate, and a shorter survival time [6,17,18]. This suggests that the overexpression of the HER2 receptor may be playing an important role in these cancers; for example, it may drive spontaneous receptor homodimerization, or heterodimerization with other HER family members, which inevitably leads to increased cell proliferation and invasiveness [19,20]. Moreover, this overexpression can lead to a 100–200-fold increase in the concentration of the HER2 protein in a tumor compared to normal tissue [7,21,22], suggesting that HER2 can serve as a tumor-associated antigen (TAA) [21,23]. TAAs differ from normal cellular proteins in their localization, levels of expression, or processing by the major histocompatibility complex (MHC), which allows for their effective targeting in tumors [24], and since the processing of the HER2 receptor will theoretically increase due to its overexpression, leading to an increase in the supply of HER2 peptides that can occupy a significant number of MHC molecules compared to other peptides, the HER2 antigen, therefore, can be effectively used as a target for cancer therapy [21].

As mentioned earlier, HER2 and HER3 are incomplete signaling molecules; nevertheless, a substantial amount of data confirms that the HER2/HER3 heterodimer represents the most active signaling dimer of the family and was found to have the highest mitogenic potential [11,25,26].

The main mechanism for HER2 overexpression in tumor cells is gene amplification [6]. However, HER2 overexpression can be observed in tumor cells both with and without gene amplification. In fact, a large body of evidence has accumulated in the past years linking HER2 overexpression to activating mutations in the HER2 gene, most of which are observed without apparent gene amplification [19,27]. Additionally, different transcriptional and post-transcriptional mechanisms can be responsible for HER2 overexpression in the absence of gene amplification [5,28]. A recent study proved that miRNAs play a key role in controlling HER2 expression. Furthermore, HSP90 was reported to influence the expression levels of HER1 and HER2 by promoting their dimerization and inhibiting their degradation via ubiquitin [29].

### HER2 Expression Status in Various Cancers

HER2 overexpression was first reported in breast cancer [16], which together with gastric cancer represents the most studied oncotype for changes in the expression levels of HER2 oncogene. Although HER2 overexpression has been reported in malignancies other than breast and gastric cancers, previous studies have mostly focused on one oncotype, making it difficult to compare HER2 overexpression across different types of cancers [30]. Several studies [19,31,32,33] offer a reasonable assessment for the HER2 overexpression status in different malignancies, but based on the criteria of Scholl et al. [34], the most accurate must be the research by Yan et al. (Table 1) [30], which provides the largest database that compares HER2 across diverse tumors with approximately 38,000 cases (the number of samples in each tumor type was greater than 50 with the exception of gliomas (low-grade), oligodendrogliomas, penile cancers, pituitary cancers, solitary fibrous tumors and testicular cancers), in addition to the advantage of using the same accredited laboratory setting for all the samples, eliminating all sources of variability.

Interestingly, HER2 gene amplification and protein expression were most frequently detected in cancers of epithelial origin and rather rarely in cancers of other tissue origin. It is also worth noting that the study of Yan et al., had some limitations. As the authors stated, there may be a sampling bias due to the fact that the study was conducted on samples submitted to a commercial molecular profiling laboratory and that only difficult cases would be submitted for HER2 expression analysis [30].

It is also worth noting that the majority of the studies for HER2-targeted therapy have focused on breast and gastric cancers and, according to the findings of Yan et al., the incidences of HER2-positivity in bladder, gallbladder, cholangiocarcinomas (extrahepatic), and esophageal and esophagogastric junction cancers were relatively high compared to breast and stomach cancers, implying that these oncotypes may also benefit from HER2-targeted therapies and that they should be considered for the clinical trials of these therapies [30].

## 4. HER2 as a Target for Therapy

The HER2 receptor is one of the most studied TAAs for cancer therapy. There are three reasons that make HER2 a valid candidate for targeted therapy: (1) Its low expression levels in normal tissues, (2) Its overexpression in a significant part of human tumors, and (3) The critical role that its overexpression appears to play in the biological development of cancer cells. Thus, cancer cell proliferation can be inhibited by interfering with the expression or activity of the HER2 receptor. [24,66]. HER2-positive cancers had a very poor prognosis prior to the introduction of HER2-targeted therapies, which have revolutionized the treatment of these cancers and significantly improved their outcome [67,68]. Studies have shown that high levels of HER2 receptor are associated with improved survival in breast cancer patients receiving HER2-targeted therapies [69].

Over the past two decades, remarkable progress has been made in the development of HER2-positive cancer treatments, with a significant number of drugs targeting the HER2 receptor moving into clinical practice, including monoclonal antibodies, inhibitors of tyrosine kinase, antibody–drug conjugates and adoptive T-cell therapies, as well as vaccines [23]. Currently, the FDA has approved several treatments against HER2, many of which have appeared in the last 3 years, with three new drugs approved in 2020 alone.

### 4.1. Antibody-Based Therapies

The introduction of passive immunotherapy with HER2-directed monoclonal antibodies (mAbs), in combination with chemotherapy, have led to improved clinical outcomes in patients, especially in the case of HER2-positive metastatic breast cancer. These improvements were largely attributed to the suppression of the oncogenic intracellular pathways triggered by HER2 [23].

Of course, mAbs are a less toxic approach to cancer treatment when compared to cytotoxic chemotherapeutic drugs. However, there were still some concerns related to the use of mAbs directed against TAAs, since these antigenic targets are also expressed on normal cells, which in turn cause various degrees of toxicity resulting from the disruption of normal cellular function in normal tissues. Fortunately, it turned out that mAbs have good toxicity profiles and are safe to use [70,71,72].

#### 4.1.1. Trastuzumab

An IgG1 humanized monoclonal antibody that binds to the extracellular domain of HER2, trastuzumab was the first anti-HER2 drug to be approved by the FDA for the treatment of metastatic breast cancer in the late 1990s, and was later found to be effective in treating early stage disease [23,67,73,74]. Administration of trastuzumab significantly improved the outcome of patients with HER2-positive breast cancer, and some studies have shown that it increased survival in approximately 85% of patients by 10 years [75].

Although the exact mechanism of action for the antitumor effect of trastuzumab is not completely understood, numerous plausible explanations have been proposed, such as the inhibition of ligand-independent dimerization, antibody-dependent cell-mediated cytotoxicity (ADCC), the inhibition of HER2 extracellular domain cleavage (preventing the formation of truncated constitutively active forms of HER2), blockade of downstream signal pathways, inhibition of angiogenesis, activation of apoptosis, induction of cell cycle arrest, and induced internalization of HER2 with consequent enhanced intracellular degradation [33]. It is also worth mentioning that trastuzumab-induced ADCC has been shown to play a very important role in the antitumor effect, since the knockout of Fc receptors in mice significantly reduced the ability of trastuzumab to stop tumor growth in vivo [76].

Despite the great leap forward achieved in clinical practice by the use of trastuzumab, as well as the improved outcomes in patients with HER2-positive breast cancer, there are still few problems limiting the use of trastuzumab, including resistance and cardiotoxicity [67,77], not to mention that the promising results of trastuzumab are limited to those patients with particularly high levels of HER2, leaving a large number of breast cancer cases, along with other types of cancers in which HER2 levels range from low to moderate, untreatable [78]. Studies on trastuzumab-resistant cell lines showed that neither HER2 itself nor downstream signaling pathways were disabled. In addition, it was found that EGFR is expressed at high levels in these cells, and it is believed that this is a mechanism for bypassing HER2 suppression and maintaining tumor growth [78].

#### 4.1.2. Pertuzumab

A fully humanized monoclonal antibody specific for a different HER2 epitope than trastuzumab, pertuzumab binds the extracellular dimerization subdomain of HER2, preventing it from interacting with other members of the HER family and thereby inhibiting tumor cell growth. Studies have also shown that pertuzumab efficiently inhibits the ligand-induced dimerization of HER2/HER3, whereas trastuzumab has only a negligible effect in the presence of a ligand [23,79]. Although pertuzumab itself exhibits only moderate antitumor activity, a synergistic effect has been demonstrated when used with trastuzumab [80,81], and as a result, the FDA approved the use of pertuzumab in combination with trastuzumab in 2012 for the treatment of patients with HER2-positive metastatic breast cancer [82]. It was later shown that pertuzumab has antitumor activity in neoadjuvant settings, where a therapy is administered prior to the main treatment to help reduce the tumor size or eliminate cancer cells that have spread [79]. It was then that the FDA approved the use of pertuzumab in the neoadjuvant setting in 2013. This was the first instance of the use of neoadjuvant treatment for breast cancer [82].

#### 4.1.3. Margetuximab

Margetuximab is a recently FDA approved chimeric human/mouse IgG1 monoclonal antibody specific for HER2 receptor. It binds to the same epitope as does trastuzumab, and is also equipped with a modified Fc region that increases affinity to the stimulating Fc receptor, FcyRIIIa (CD16A), expressed by macrophages and natural killer cells, while reducing affinity to the inhibitory Fc receptor, FcyRIIB (CD32B) [83,84]. In 2020, the FDA approved the use of margetuximab in combination with chemotherapy for the treatment of adult patients with HER2-positive metastatic breast cancer [85]. Introducing a combination of chemotherapy and passive immunotherapy with HER2-directed mAbs leads to improved clinical outcomes in patients, especially in the case of HER2-positive metastatic breast cancer. These improvements were largely attributed to the suppression of carcinogenic intracellular pathways activated by HER2 [23]. Furthermore, margetuximab proved to be more effective than trastuzumab against HER2-positive cancer cells expressing low levels of HER2, or that are resistant to the antiproliferative activity of trastuzumab, as well as in patients with the low-affinity variant of the FcyRIIIa receptor (158F, homozygous or heterozygous), as shown by antibody-dependent cytotoxicity assays [86].

### 4.2. Tyrosine Kinase Inhibitors

In addition to targeting HER2 with monoclonal antibodies, there is another approach based on small molecule tyrosine kinase inhibitors (TKIs) that compete with ATP molecules when binding to the catalytic kinase site of HER2, which inhibits tyrosine kinase phosphorylation and suppresses subsequent signaling. The understanding of HER2 signaling capability and tyrosine kinase activity prompted the development of this class of treatment [87,88]. An additional incentive for the development of TKIs was the need for therapeutic agents that could pass the blood–brain barrier to treat metastases in the central nervous system, which was increasingly observed in breast cancer patients receiving trastuzumab or other antibody-based treatments targeting HER2. Theoretically, TKIs may be more effective for treating brain metastases than antibodies, since they have a smaller molecular weight and can penetrate the blood–brain barrier more readily [68,89]. The benefits of TKIs also include oral administration and reduced cardiac toxicity, which means that patients who have developed congestive heart failure as a result of trastuzumab can be safely treated with TKIs [5,88]. Moreover, recent evidence from mouse models suggest that the antitumor efficacy of oral TKIs may be related not only to the direct inhibition of signaling through the HER2 receptor, but also to the induction of the Th1 cytotoxic immune response [23].

Until 2017, lapatinib was the only HER2-specific TKI that had received FDA approval. Several additional new TKIs targeting the HER family, including neratinib and most recently tucatinib, are now available for therapy [88].

#### 4.2.1. Lapatinib

The first TKI developed to target HER2 was lapatinib ditosylate, which restricts the phosphorylation of both HER1 and HER2 by reversibly and competitively occupying the ATP-binding sites of the intracellular kinase region and thus interfering with downstream signaling, which, in turn, leads to the induction of apoptosis and restriction of the growth and migration of cancer cells overexpressing HER2. In 2007, the FDA approved the use of lapatinib in combination with the chemotherapeutic drug capecitabine to treat patients suffering from metastatic breast cancer overexpressing HER2, who have previously received anthracycline, taxan, and trastuzumab [5,90,91,92]. However, the results of many subsequent clinical trials have shown that Lapatinib is associated with increased hepatic and cardiac toxicity, and with limited improvement in outcome. Therefore, this drug is not currently used in neoadjuvant or adjuvant settings (when a therapy is given in addition to the primary or initial therapy to maximize its effectiveness) [91].

#### 4.2.2. Neratinib

Neratinib is a second-generation TKI that irreversibly inhibits HER1, HER2, and HER4. Preclinical studies have shown that neratinib promotes cell cycle arrest in the G1-S phase and reduces the proliferation of HER2-positive cells [91,92]. Additionally, neratinib can induce HER2 downregulation through the dissociation of HSP90 and subsequent induction of ubiquitylation and endocytic degradation. Furthermore, there is some evidence that neratinib can inhibit the ATP-binding cassette transporter which might help overcome the multidrug resistance of cancer cells [92]. In 2017, the FDA approved neratinib as an extended adjuvant treatment for patients with early stage HER2-overexpressing breast cancer, after surgery and trastuzumab-based adjuvant therapy [92]. However, due to the high incidence of diarrhea, the use of neratinib is often limited in clinical practice [91].

#### 4.2.3. Tucatinib

Tucatinib is a potent, highly selective and reversible HER2 inhibitor that is approximately 1000 times more efficient against HER2 than EGFR. Due to its strong HER2 selectivity, tucatinib has fewer EGFR-associated toxic effects, such as diarrhea and rashes, which are common with many other anti-HER2 TKIs [89,91,93]. Another quality that distinguishes tucatinib is its ability to effectively cross the blood–brain barrier [68]. That being said, xenograft models showed that tucatinib was more effective against breast cancer than lapatinib, and that the addition of trastuzumab improved the tucatinib response. In addition, tucatinib was associated with improved survival compared to lapatinib or neratinib in mouse models of intracranial diseases [68,91]. In 2020, the FDA approved the use of tucatinib in combination with capecitabine and trastuzumab as a treatment for patients suffering from metastatic or advanced unresectable HER2-positive breast cancer, or those patients with brain metastases, who have previously received one or more anti-HER2-based treatments in the metastatic settings [94].

### 4.3. Antibody–Drug Conjugates (ADCs)

Since the main drawback of conventional cancer chemotherapy is that it exposes healthy, non-tumor cells to cytotoxic chemicals which results in dose-limiting toxicities, ADCs represent a promising class of treatment that provides a broader therapeutic window due to the more effective and specific technique of drug delivery. To increase tumor selectivity and minimize harm to healthy cells, ADCs use the target selectivity of monoclonal antibodies to specifically deliver cytotoxic elements to those cells that express a given antigen. With that being said, ADCs can be divided into two basic categories, cleavable and non-cleavable, based on the stability of the bond between the antibody and the chemotherapeutic payload. The main difference between them is that non-cleavable ADCs require intracellular degradation before the drug can be released and function properly. It is also worth mentioning that ADCs represent a promising treatment modality for HER2-positive cancers because of their intriguing mechanism that encompasses a potential bystander cytotoxic effect, since a significant portion of HER2-positive tumors have intra-tumor heterogeneity regarding the expression levels of HER2, and permeable cytotoxic payloads of ADCs can have cytotoxic effects on neighboring cells that do not have high HER2 expression [23,68,89].

#### 4.3.1. Trastuzumab Emtansine (T-DM1)

T-DM1 is a second-generation ADC, which is an alternative strategy to increase the efficacy of trastuzumab in cases of low HER2 expression or resistance to trastuzumab by binding it to the cellular cytotoxic agent mertansine (DM1, a potent microtubule inhibitor). The bond between trastuzumab and DM1 is not cleavable, which means that the potency of T-DM1 depends on the effective internalization of the drug and on lysosomes properly cleaving the linker of the payload [5,23,68,95].

T-DM1 was the first ADC to be approved by the FDA for the treatment of solid tumors in 2013, which completely changed the field of ADCs. It was adopted as an adjuvant (postoperative) treatment for patients with early HER2-positive breast cancer who were previously treated with taxane and trastuzumab, either separately or in combination. T-DM1 was subsequently approved in 2019 for the treatment of early HER2-positive breast cancer after neoadjuvant taxane and trastuzumab-based treatment [96].

All in all, there are numerous factors that can contribute to the toxicity of ADCs, such as payloads, mAbs, off-target effects, complement-dependent cytotoxicity (CDC), and ADCC. Therefore, the side effects of ADC are diverse. The most common side effects reported in T-DM1 were anemia, thrombocytopenia, and elevated serum concentrations of alanine aminotransferase (ALT) and aspartate aminotransferase (AST) [23,97].

#### 4.3.2. Trastuzumab Deruxtecan (T-DXd)

T-DXd is a novel ADC consisting of trastuzumab and a payload of topoisomerase I (an exatecan derivative). Due to the cleavable linker that attaches the cytotoxic payload to the antibody, a portion of the payload is released extracellularly, causing a bystander cytotoxic effect on nearby tumor cells. In addition, compared to T-DM1, the payload of T-DXd is characterized by a greater membrane permeability, which enable its release into the intercellular space and contributes to the bystander cytotoxic effect. Nonetheless, T-DXd has a short half-life, which may reduce its systemic exposure and potential side effects [23,68,88].

In preclinical studies, T-DXd has demonstrated broader antitumor activity compared to T-DM1, including efficacy against tumors with low HER2 expression. Moreover, it was also shown that T-DXd displayed activity against T-DM1-insensitive HER2-positive breast cancer cell lines. All of this resulted in T-DXd receiving accelerated FDA approval in 2019 for the treatment of adult patients with inoperable or metastatic HER2-positive breast cancer or metastatic breast cancer who had previously received two or more anti-HER2-based treatment regimens. In addition, the FDA granted this drug breakthrough therapy status in 2020 for treating patients suffering from metastatic non-small cell lung cancer (NSCLC) with a HER2 mutation after platinum-based therapy, as well as priority review for the treatment of HER2-positive metastatic adenocarcinoma of the stomach or gastroesophageal junction [23,68,96].

Unfortunately, compared to T-DM1, T-DXd has more side effects. The increases in AST or ALT levels in T-DXd are relatively rare, but hematological toxicities such as anemia, neutropenia and leukopenia, not to mention pneumonitis and interstitial lung disease, are more problematic. On the other hand, the T-DXd-induced peripheral neuropathy is less serious than the one induced by T-DM1 [97,98].

Mechanisms of action of all the above-mentioned anti-HER2 therapies are summarized in Figure 1.

#### 4.3.3. SYD985

SYD985 is a third-generation ADC consisting of trastuzumab and a duocarmycin payload, which is a powerful DNA alkylating agent, attaching to it with a cleavable linker. Endosomal proteases cleave the linker in SYD985, allowing the membrane-permeable active toxin to be released. The toxin then binds to the minor groove of DNA, causing irreversible alkylation, which can cause cell death in both dividing and non-dividing cells in the tumor microenvironment, as well as in neighboring tumor cells causing a bystander effect [23,89].

Both SYD985 and T-DM1 were evaluated for their anticancer efficacy in vitro using HER2 overexpressing cell lines, and the results revealed that both of them have comparable inhibitory activity. Moreover, SYD985 was investigated in vivo on xenograft models obtained from cell lines and breast cancer patients with different HER2 expression levels. SYD985 outperformed TDM1 in terms of anticancer activity and was 3–50 times more cytotoxic than T-DM1 in cell lines with low HER2 expression. It is also worth mentioning that SYD985 has been shown to be effective in T-DM1-resistant cell lines. Interestingly, several clinical trials have also reported that SYD985 is effective in patients with HER2-low metastatic breast cancer. The findings from a dose-escalation phase I study revealed promising results in heavily pretreated patients and prompted the FDA to grant SYD985 a Fast Track designation in 2018 [23,68,89,99].

### 4.4. Adoptive T-Cell Therapies

Immunotherapeutic strategies against HER2-expressing tumors have been traditionally based on passive immunotherapy, such as the HER2-blocking mAbs. This passive approach has proven to be effective in the treatment of HER2-positive breast cancer. However, allergic and hypersensitivity reactions, a resistance to mAb, a lack of immunological memory, and the relapse of the tumor are still considered important problems in clinical practice [7,100,101]. It has been a matter of debate whether an endogenous T cell response can contribute to tumor regression, since tumor development is often reported even in the presence of significant amounts of blood-circulating or tumor-infiltrating T cells. It was only at the beginning of the 21st century that it became obvious that the presence of intratumoral T-lymphocytes was associated with a favorable prognosis of tumors [102]. Inspired by the unprecedented advances in cancer immunotherapy over the past decade, such as the success of immune checkpoint inhibitors (ICIs) and adoptive cell therapy with chimeric antigen receptor T cells, which revolutionized the treatment of hematological malignancies, many studies have started to use this “active immunotherapy” in solid tumors, but unfortunately, this method of treatment has not yet proved its sufficiency to achieve significant antitumor activity in these tumors [103].

Adoptive cell therapy (ACT) involves the isolation of the patient’s own immune cells, followed by their ex vivo expansion and manipulation before reinfusing them back into the patient. In most cases, T cells isolated from tumors or peripheral blood are used for this purpose, but other subsets of immune cells, including natural killer cells, can also be used [104]. Adoptive T-cell therapy comprises: (a) genetically modifying T-Cell Receptors (TCRs) that are able to recognize, with high affinity, antigens presented by MHC I; (b) generating chimeric antigen receptor T cells (CARs) by fusing the natural T-cell receptor to a selective antibody; (c) infusing ex vivo-expanded tumor-infiltrating lymphocytes or ex vivo-tumor antigen-primed peripheral blood T lymphocytes [23].

The essence of this approach is basically an attempt to overcome tumor tolerance by collecting the potentially tumor-reactive T cells from the patient and amplifying them ex vivo, or generating these cells ex vivo from peripheral blood T lymphocytes for them to be eventually given back to the patient in order to help the body fight the tumor [102]. Both TCR and CAR gene transfer adoptive T-cell therapy generate polyclonal T cells with activities against TAAs that are naturally absent, therefore providing a versatile and highly subtle tool for cancer therapy [101]. Another advantage of using adoptive T-cell therapy is that it can act as “live drug”, taking into account the ability of these infused cells to reproduce, engraft, and survive in vivo, providing adaptability and enabling long-term remissions [104].

#### 4.4.1. CAR-T-Cell Therapy

With the discovery that the intracellular CD3ξ subunit of the T cell receptor is sufficient for transmitting signals that can activate the cell, many attempts were made to create a chimeric antigen receptor by replacing the extracellular α and β chains of the TCR with a single-chain variant fragment (scFv) of immunoglobulin. Unlike TCR, the extracellular antigen recognition domain in CAR is covalently bound to the intracellular CD3ξ. These “first generation” CAR constructs demonstrated limited activity in vivo due to the poor persistence of T cells, which led to the addition of single (second generation) or multiple (third generation) costimulatory domains (CD28, 4-1BB and OX40…), which help increase the efficiency and delay the exhaustion of the cells, hence improving the antitumor activity of the cells. It is also worth noting that the particular molecular design of the CAR construct is a factor that greatly affects the potency of the developed cellular product [103,104].

There are several advantages of CAR-T-cell therapy compared to TCRs. To begin, target antigen recognition occurs in an MHC-independent manner and, therefore, does not require tissue compatibility, which facilitates product development and helps to overcome MHC downregulation as a mechanism for tumor escape. Moreover, in addition to peptide antigens, CAR-T cells can recognize carbohydrate and glycolipid antigens, not to mention that CARs recognize antigens in multiple dimensions, which increases the pool of targetable cancer antigens. Yet, cognate antigens are consequently restricted to surface molecules [101,103,105].

Although CAR-T cells have shown promising results in treating hematological malignancies, they are not as effective in the treatment of solid tumors and often have toxic side effects. This might be attributed to a number of reasons; for example, the high heterogeneity of solid tumors significantly complicates their treatment with CAR-T-cell therapy. In addition, cells of the hematological tumor are scattered while solid tumors tend to form solid masses, which not only makes the homing of CAR-T cells from peripheral blood to the tumor site challenging, but also results in a buildup of several types of immunosuppressive cells and molecules. Moreover, in most cases, antigens of hematological tumors are specific and are not expressed in other healthy tissues (tumor-specific antigens, TSAs), whereas antigens of solid tumors are usually present in small amounts in other healthy tissues (TAAs), resulting in mistargeted effects referred to as “on-target, off-tumor” toxicity [88,101,105]. Another related problem is the downregulation of the targeted antigen since it is usually not essential for tumor cell survival and can be replaced by another protein, as confirmed by some studies where a rapid downregulation of the targeted antigen was observed after treatment [103].

The side effects associated with CAR-T cells are directly related to their mechanism of action. Upon recognition of the target antigen, the CAR binds to it and transmits an activation signal with the help of co-stimulatory molecules. Cytokines and chemokines are secreted to activate additional components of the immune system and help recruit them to the tumor site, as well as to kill the target cells. The problem is that CAR-T cells are designed in such a way that they all target the same epitope; therefore, when activated, they overload the system with a massive signal. Cytokine and chemokine overproduction can lead to what is known as cytokine release syndrome (CRS), a sepsis-like condition that needs to be carefully managed, otherwise it may cause death. In addition, these secreted cytokines and chemokines can trigger a strong activation of monocytes and macrophages which can pass the blood–brain barrier. As a result of this, the central nervous system might enter a CAR-T-induced encephalopathic state. Thus, a significant amount of current research is aimed at optimizing the supportive care for patients receiving CAR-T-cell therapy and developing CAR-T with lower toxicity [106].

The first evidence of the clinical use of CAR-T cells against HER2-positive tumors dates back to 2010, when a patient with metastatic HER2-positive colon cancer received HER2-CAR-T cells of the third generation. Within 15 min of the administration, the patient began experiencing multiple cardiac arrests and respiratory failure, and eventually died 5 days later. Authors speculated that this effect was caused by the recognition of the low constitutive expression of HER2 in normal lung epithelial tissue. Laboratory analysis was consistent with CRS. However, the low levels of HER2 present in normal lung tissue are unlikely to cause the lung toxicity observed with HER2-CAR-T cells, since lung toxicity has not been observed with other HER2-directed therapies, such as trastuzumab, pertuzumab, T-DM1, and anti-HER2 TKI [88,101,103]. Here, a very important question arises: why it is relatively safe to target TAAs with mAbs, but at the same time using the same TAAs as targets for adoptive T-cell therapy turns out to be highly toxic!?

However, the situation has changed drastically since then, and the results of a number of recent studies are promising for the use of CAR-T cells in HER2-positive tumors [88]. Currently, there are many clinical trials that are being conducted to evaluate the effectiveness of CAR-T cells as a treatment of HER2-positive tumors (Table 2), but up until this moment, there is still no approved CAR-T-cell therapy for HER2-positive tumors.

#### 4.4.2. TCR-T-Cell Therapy

The challenge with using CAR-T cells to treat solid and hematological tumors is that the targeted antigens should be presented on the cell surface [105]. An alternative approach for adoptive T-cell therapy is to use modified TCRs, as there are several potential advantages for using them in therapy. First of all, since MHC molecules can represent peptide chains obtained from the cell surface or intracellular proteins, the diversity of targetable antigens is much greater with TCRs. In addition, using TCRs, it is possible to target mutated cancer-associated proteins, thereby increasing the selectivity of the engineered cells. However, this approach also has some drawbacks, such as MHC restriction, which limits the application of TCR therapy to specific subtypes of HLA. In addition, the risk of cross-reactivity with another antigen is of high concern [103].

Some studies have shown that TCR-T cells have greater sensitivity compared to CAR-T cells, even when the CARs were expressed at a higher density on the cell surface. At the same time, TCR-T cells were found to secrete lower levels of cytokines. Thus, TCR-T-cell therapy has the advantage of a lower risk of cytokine release syndrome compared to a CAR-T-cell approach and appears to be moving forward in the race for adoptive T-cell therapy in solid tumors [103]. It was also found that CAR-T cells are more potent effector cells, killing tumor cells more efficiently than TCR-T cells in a shorter period of time. However, an increase in the antigen exposure significantly disrupts the expansion of CAR-T cells, a phenotype characterized by the increased expression of coinhibitory molecules. On the contrary, TCR-T cells proved to be better at high antigenic pressure, maintaining stable expansion rates with a lower expression of coinhibitory molecules and a comparable clearance of tumor cells [107]. Research is still ongoing, but up until this moment, there is still no approved TCR-T-cell therapy for HER2-positive tumors.

Simply put, the low efficiency of T-cell immunotherapy in solid tumors, along with the necessity to bypass the limitations of MHC restriction of conventional TCRs, emphasizes the need for developing additional immunotherapeutic agents, in particular NK-cell-based therapies. Human NK cells are considered the main innate immune effector cells in response to tumors. Recently, the development of CAR-NK cells for the treatment of solid tumors has gained a great deal of interest. CAR-NK cells have some advantages over CAR-T cells, such as CRS being less likely to develop using this approach as NK cells produce less cytokines compared to T lymphocytes, as well as the feasibility for “off-the-shelf” production. Thus, the development of HER2-specific CAR-NK cells may contribute to more desirable outcomes, providing a superior safety and efficacy profile compared to CAR/TCR-T cells [108].

### 4.5. Vaccines

Many studies have reported that cellular and/or humoral immune responses against HER2 are observed in patients suffering from HER2-overexpressing breast cancer. These observations, together with reports on the effectiveness of mAbs-mediated passive immunotherapy against HER2, suggest that the stimulation of anti-HER2 immune response via vaccination can be used for the treatment of HER2-positive cancers or to prevent disease recurrence [100,109,110]. Such an approach was encouraged by the high immunogenicity of the HER2 antigen [88].

Despite the huge success achieved by mAbs in treating HER2-positive tumors, their use in therapy has many disadvantages, including the development of resistance, the short half-life, and the lack of immunological memory, all of which result in disease recurrence, continuous application, and only a temporary control of the disease [111,112]. Fortunately, vaccination has the potential to overcome these disadvantages by stimulating the patient’s immune system to produce an antitumor immune response that includes both cellular and the humoral immunity, i.e., using the body’s own immune system to detect and eliminate cancer cells. In this case, the immune system of the patient will produce its own antibodies [7,100]. Vaccination is considered as a middle solution between mAbs and adoptive cell therapy as it produces less cytotoxicity compared to ACT while also being more effective than mAbs-mediated immunotherapy, since it requires fewer administrations, is more cost-effective, and generates immunological memory, which can help the immune system detect antigens and respond to them upon future exposures, protecting against tumor recurrence [109,113,114].

One of the bases of active immunotherapy via vaccination is the ability of the immune system to discriminate self-antigens that are normally expressed on the surface of healthy cells from those that are abnormally overexpressed on tumor cells. In fact, HER2 is the most thoroughly studied TAA for breast cancer vaccines [70] Both CD8+ and CD4+ T cells participate in the induction of immune responses against HER2-positive cancers. CD8+ T cells are able to recognize HER2 peptides represented on MCH class I molecules, trigger apoptosis and cell cycle arrest, as well as eliminate tumor cells by releasing IFN-γ, TNF-α and a number of other cytotoxic cytokines. CD4+ T cells are also crucial players in antitumor immunity, as they are essential for the development of humoral immune response [100,115]. However, as stated earlier, the HER2 antigen is expressed in normal tissues, which means that there is a natural immunological tolerance to it, a fact that poses a significant barrier to successful vaccination against this oncoprotein. Accordingly, an effective vaccine is one that induces the right conditions to overcome the immune tolerance without causing any autoimmune side effects [109].

There are several different approaches that can be utilized to develop a vaccine, which is a considerable advantage for using vaccines in treatment, since these different approaches provide many options, making vaccination adjustable and suitable for any situation. These approaches include B- and T-cell peptide-based vaccines and dendritic cell-based vaccines [116,117].

#### 4.5.1. T-Cell Peptide Vaccines

This category consists of peptides from various parts of the HER2 molecule, including E75, GP2, and AE37. Peptide-based vaccines represent the most studied strategy for developing anti-HER2 vaccines [109].

1.E75:

E75 is a nine-amino acid peptide derived from the extracellular domain (ECD) of the HER2 molecule (amino acids 369–377: KIFGSLAFL). The peptide is an immunodominant epitope of cytotoxic T lymphocytes (CTLs) with a high affinity for HLA-A2 and HLA-A3 molecules and is the most studied candidate for a peptide vaccine against HER2. Nelipepimut-S (or NeuVax) is the vaccine developed against this peptide, and it is the only HER2-vaccine that is being investigated in a phase III trial (PRESENT trial, NCT01479244) [7,118,119].

The results of phase I/II studies have shown that NeuVax, (with GM-CSF as an adjuvant), is well tolerated and is able to effectively induce HER2-specific immunity in patients with HER2-positive breast cancer, not to mention that the vaccine was proven to reduce the recurrence rate when used in an adjuvant setting. Interestingly, some studies reported that patients with low-HER2-expressing tumors had more robust immune responses and, as a result, might benefit the most from vaccination [109,120]. It is also worth mentioning that some preclinical findings have suggested that the pre-treatment of HER2-overexpressing breast cancer cells with trastuzumab before the administration of E75 vaccines resulted in greater expansion and increased specific cytotoxicity of stimulated CTLs, suggesting that a combination of passive and active immunotherapy may have a synergistic effect. These findings can be explained by the increased internalization and faster processing rates of HER2 following trastuzumab administration, which, in turn, leads to the enhanced presentation of HER2-specific epitopes on MHC I molecules. This means that trastuzumab can be used as vaccine-potentiating agent [100,109]. However, the results of the phase III clinical trial reported no significant difference between NeuVax and a placebo in terms of the relapse-free survival of patients [120], but it should be emphasized that the vaccine in this study was used without pretreatment with trastuzumab, which should be taken into account in future studies.

2.GP2

GP2 is another nine-amino acid peptide derived from the transmembrane domain (TMD) of the HER2 molecule (amino acids 654–662: IISAVVGIL), which has been known to have a low affinity for HLA-A2, which makes it an unsuitable option for vaccine development. However, multiple in vitro and phase I and II studies have reported that GP2 is as effective as E75 in terms of inducing an HER2-specific immune response [100,121], making GP2 a promising antigen for an HER2-positive cancer vaccine application. In addition, a phase II clinical trial has shown that a combination of pretreatment with trastuzumab and subsequent GP2 vaccination has possible clinical benefits [122].

3.AE37

The short-lived response is considered one of the important hurdles that the peptide-based vaccination approach faces, probably due to the lack of activation of other components of the immune system, such as CD4+ T-helper cells. In addition, both previous vaccines have the limitation of being restricted to HLA class I, which reduces their ability to stimulate the immune system more broadly and minimizes the population of patients in whom they can be used [23,123]. Another challenge for HER2 vaccines is the downregulation of MHC class I molecules as a tumor-escape mechanism. This phenomenon impairs the ability to present peptide ligands to lymphocytes, allowing the tumor to avoid immunosurveillance [109,123].

The above-mentioned facts emphasize the need for a HER2 vaccine capable of inducing a complex immune response, including CTLs, antibodies, and CD4+ T cells. This is where AE37 comes into play with its unique design that enables the activation of both CD8+ and CD4+ cells. AE37 is a modified version of the naturally occurring MHC class I peptide AE36, which is derived from the intracellular domain (ICD) of HER2 (amino acids 776–790: GVGSPYVSRLLGICL). AE36 was fused at the C-terminal end with the Ii-Key, a sequence of four amino acids: LRMK. This sequence can interact with the MHC-II molecules which increases the potency of the vaccine to activate CD4+ T cells. It is also worth noting that the results of a phase I study proved the safety of the vaccine in addition to its ability to induce HER2-specific immune responses, even when no adjuvant is used, which is the first case in the history of peptide vaccines [7,100,109,124].

#### 4.5.2. B-Cell Peptide Vaccines

The tremendous success of passive immunotherapy using HER2-targeted mAbs has raised great interest in the search for new B-cell epitopes (the antibody-binding part of the antigen) to be used as targets for anti-HER2 vaccines, especially as the role of vaccination in the management of HER2-positive cancers is expanding [7,125]. This was further reinforced by recognizing the critical role of B cells in tumor immunity [126].

Active immunotherapy with B-cell epitopes makes it possible to generate a stable immune response by presenting the B-cell epitopes to the patient’s immune system in order to activate its own antitumor response, thus providing a continuous supply of antibodies against HER2 and forming immunological memory [111,127]. The rationale for this approach is the fact that antibodies can bind to tumor antigens expressed on the cell surface without the involvement of MHC-I. Thus, MHC-I downregulation as a tumor-evasion mechanism does not affect the availability of epitopes for antibodies, not to mention that this will allow for a wider usage by all patients regardless of their HLA type [128].

Various B-cell epitopes from HER2 have been identified and are being tested for vaccine development.

1.HER-Vaxx:

Unlike monoclonal antibodies, that target only a single epitope, HER-Vaxx consists of three immunodominant B-cell epitopes derived from ECD of HER2 using computer algorithms (P4: amino acids 378–394 PESFDGDPASNTAPLQP; P6: amino acids 545–560 RVLQGLPREYVNARHC; and P7: amino acids 610–623 YMPIWKFPDEEGAC) [129,130]. Although these epitopes showed high immunogenicity, as well as an excellent safety profile when used individually in a phase I study, there were still some drawbacks related to their stability and solubility [131], which led to their fusion into one hybrid peptide P467 (PESFDGDPASNTAPLQPRVLQGLPREYVNARHSLPYMPIWKFPDEEGAC). P467 was additionally coupled to CRM197, a carrier protein that has the ability to rapidly activate CD4 T-cells with a mixed Th1 and Th2 cytokine profile, hence inducing B cell activation. It is also worth mentioning that his vaccine also uses the Th1/Th2-stimulating adjuvant Montanid [129].

Results from a phase I study reported that HER-Vaxx was well tolerated, safe, and activated innate and adaptive immunity and inducing HER2-specific antibodies [111,132]. In addition, Tobias et al., reported a significant increase in IFNγ-producing CD8+ T cells following vaccination with HER-Vaxx in vivo, indicating that the vaccine induced cellular responses. According to the authors, this effect may be due to Montanide-driven bystander activation [129]. Currently, two phase II studies are ongoing to evaluate the effectiveness of HER-Vaxx in HER2-positive cancers [112,133]. The preliminary results show that the antibodies induced by HER-Vaxx are comparable to trastuzumab [132].

2.B-Vaxx

The development of B-Vaxx took a more rational path, since it was based on the binding sites of pertuzumab and trastuzumab, which were already proven to be functional and capable of inhibiting HER2 signaling. Using X-ray structures of HER2-pertuzumab and HER2-trastuzumab complexes, it was found that the binding site of pertuzumab encompasses the residues (266–333), and the binding site of trastuzumab encompasses the residues (563–626). Following intensive research to identify the most effective sequence in each of these sites, the epitope (266–296) within the pertuzumab binding site and the epitope (597–626) within the trastuzumab binding site were designated as the most promising epitopes for further use in vaccination, as they demonstrated the highest affinity and titer antibody responses. These two epitopes, in combination with a potent adjuvant that is known as nor-muramyl dipeptide (n-MDP), constitute the B-Vaxx vaccine [7,127,134,135]. Results from a phase I study with B-Vaxx showed that the vaccine is well tolerated and effectively induces humoral responses in a large proportion of patients [136,137].

#### 4.5.3. Dendritic Cell-Based Vaccines

Dendritic cells (DCs) are a special type of immune cells that are highly specialized in antigen presentation. They have the ability to process antigens and migrate to lymph nodes, where they can present these antigens as peptides to naive T cells with the help of MHC I and II molecules. Furthermore, DCs play a crucial role in regulating antibody-based responses, since they are able to interact directly with B cells and induce proliferation and differentiation of CD4+ T cells. All the aforementioned characteristics make DC-based vaccination one of the most relevant approaches for tumor vaccine development [100,138,139].

There are many approaches for using DCs as a cancer vaccine, the majority of which involve the ex vivo generation of TAA-loaded DCs or transfected DC-expressing tumor antigens, which are subsequently administered as vaccines [138]. These approaches involve isolating inactive or immature dendritic cells (iDC) from the patient’s peripheral blood, which are then either (I) pulsed with recombinant TAAs, (II) transfected with DNA/RNA encoding TAAs, or even bulk RNA extracted from tumor cells, or (III) fused with tumor cells acquired from the patient to obtain fused cells. The antigen-loaded iDCs are then stimulated by specific cytokines to maturate before injecting them back into the patient, where the mature DCs can present cancer antigens to CD4+/CD8+ T cells, inducing a sustained antitumor response [139,140,141,142,143].

According to some reports, DC-based vaccination has proven to be more effective in inducing cellular immunity compared to peptide-based vaccination [100]. Moreover, the great advantage of this approach is the lack of side effects when autologous DCs are used for the production of vaccines [140], not to mention that it allows for the circumvention of the malfunction of endogenously activated DCs, which occurs in many cancer patients, as well as for the transfer of highly active induced cells that are able to enhance the immune response against tumor cells [143]. In several phase I clinical trials, promising results were obtained regarding the use of DCs in cancer immunotherapy, where this vaccination approach proved to be safe, tolerable, and capable of enhancing the infiltration of lymphocytes into the tumor site and inducing the specific activation of cellular and humoral reactions. Phase II clinical trials are currently underway, testing the effectiveness of DC-based vaccines in HER2-positive tumors [139,141,142]. It is also worth noting that the cell fusion vaccine has demonstrated a strong antitumor response, since it provides a wide range of antigens capable of electing strong immunogenic responses [139,140].

## 5. Mechanisms of Resistance to HER2-Targeted Therapies

HER2-targeted therapies have undeniably improved the prognosis of HER2-positive cancer patients. However, and despite all the advancements in this field, many patients still suffer from recurrences and disease progression on treatment, suggesting that tumors can develop resistance to these therapies. Furthermore, a growing body of evidence shows that the vast majority of patients do not respond to these therapies, and those who initially respond eventually develop resistance to treatment [144,145,146]. This is particularly relevant in the case of trastuzumab, as studies have indicated that the majority of patients who respond to treatment based on trastuzumab acquire resistance within a year. Furthermore, the numbers revealed that around 70% of HER2-positive breast cancer cases have some form of resistance to trastuzumab [145,147].

Since HER2 is involved in a range of biological processes such as proliferation, survival, angiogenesis, and metastasis, resistance to HER2-targeted therapies can develop due to various mechanisms. A deeper understanding of these mechanisms, as well as the possible strategies to overcome them, will help improve the outcomes for patients with HER2-positive cancers [144,147]. Table 3 outlines the mechanisms of resistance to HER2-targeted therapies with the suggested strategies to overcome it.

### Tumor MicroEnvironment (TME)

TME is responsible for a significant portion of the resistance to targeted therapies in solid tumors, especially in the case of adoptive T-cell therapies that have generally failed, mainly owing to the inhibitory effect of the TME, because antigen-specific T cells must obtain access to, and function within, the TME in order to successfully detect and attack tumor cells in vivo. The TME of solid tumors develops an aria of immunosuppression and inflammation surrounding the tumor, promoting tumor growth and protecting it from the immune system. Studies have revealed that the immuno-inhibitory effect of TME can be explained by two reasons: inability of T-cells to home in on the tumor site due to the low levels of chemokines for migration and dominant suppression via immune inhibitory pathways [154,155]. One of the proposed solutions to overcome the TME inhibition is by combining two separate targeted cell treatments, one targeting TME components and the other targeting tumor cells. Rodriguez-Garcia et al., recently proved the efficacy of such an approach by sequentially administering TAM targeting T-cells followed by cancer targeting T-cells, which resulted in tumor regression and extended survival in mouse models [155].

In line with that and since the immune check point inhibitors (ICIs) have achieved great success in the treatment of tumors, it is only logical to use a combination of HER2-targeted therapies and ICIs for the treatment of resistant tumors as the ICIs will help overcome the immuno-inhibitory effect of the TME. In fact, in preclinical models, ICIs have been shown to improve the therapeutic efficacy of HER2-targeted therapies [144,156]. However, in clinical studies, the benefits of combining ICIs with HER2-targeted therapies have been modest, not to mention that this treatment modality was associated with increased toxicities. Nonetheless, certain clinical trials are being conducted to determine whether patients with advanced HER2-positive breast cancer can benefit from ICIs treatment (NCT04740918 and NCT03199885) [144].

## 6. Future Perspective and Conclusions

Over the past ten years, significant progress has been made in the treatment of HER2-positive tumors, progress that has the potential to drastically alter the shape of cancer treatment. In line with that, a group of scientists found a solution to the manufacturing challenges, frequent dosing, and the high cost of antibody production by using an in-vitro-transcribed mRNA (IVT-mRNA) system for endogenous protein expression, in which they deliver trastuzumab mRNA to the liver so that the patient could produce their own antibodies. One of the advantages of this system is that it allows for the better control of posttranslational modifications on the antibody. In comparison with DNA plasmids and viral vectors, the IVT-mRNA system has demonstrated the ability to generate higher levels of circulating antibodies in vivo. In addition, the authors conducted pharmacodynamic and pharmacokinetic studies examining the antitumor activity of the expressed antibodies and showed that the use of IVT-mRNA for the expression of a full-size therapeutic antibody trastuzumab in the liver is an effective strategy for cancer treatment [29].

Another breakthrough was achieved by identifying a new HER2-negative subtype of breast cancer sensitive to anti-HER2 therapy. A group of scientists reported that HER2-signaling functional activity is better than HER2 protein quantification to identify patients eligible for HER2-targeted therapy. The authors tested HER2-negative patients with a HER2-signaling function (CELx HSF) test, which uses a biosensor to examine the patient’s living tumor cells for the presence of unusually high levels of HER2-related signaling (HSFs+), to determine if they can benefit from anti-HER2 therapies. Their findings revealed that 20–25% of 114 patients suffering from what has been classified as HER2-negative breast cancer have abnormal HER2 signaling and even responded to five approved anti-HER2 therapies (Pertuzumab, Trastuzumab, Lapatinib, Afatinib, and Neratinib). This suggests that there is a good chance of treating a significant proportion of patients with tumors that have been classified as HER2-negative by HER2-targeted therapies [157].

Another interesting approach for the development of adoptive T-cell therapy is to design the cells to require combinatorial antigen in order to be activated. Roybal et al., have developed a combinatorically activated T-cell circuit in which a synthetic Notch receptor for one antigen triggers the production of a CAR for another. This means that the cells initially express a synthetic Notch receptor targeting a specific antigen (for example, a TAA). As this receptor becomes activated, it triggers the expression of a CAR receptor targeting another antigen (another TAA). As a result, only in the presence of tumor cells expressing both antigens are these dual-receptor T cells activated, increasing selectivity and reducing the on-target off-tumor toxicity of these therapies [158].

In conclusion, we can remark on the importance of the HER2 molecule as an antigen and target for the development of cancer therapy, since the implementation of trastuzumab in the treatment of HER2 breast cancer, which significantly alters the treatment of patients with HER2-positive tumors, has revolutionized this field by changing the entire prognosis of this cancer subtype, up to the present moment, with all the developments and novel treatments targeting this molecule. However, we are still in need of a deeper understanding of the molecular biology underlying HER2-positive cancers, as well as the molecular mechanisms responsible for developing resistance to HER2-targeted therapies, so that we can develop more suited and tailored therapies.

## Figures and Tables

**Figure 1 cancers-14-06173-f001:**
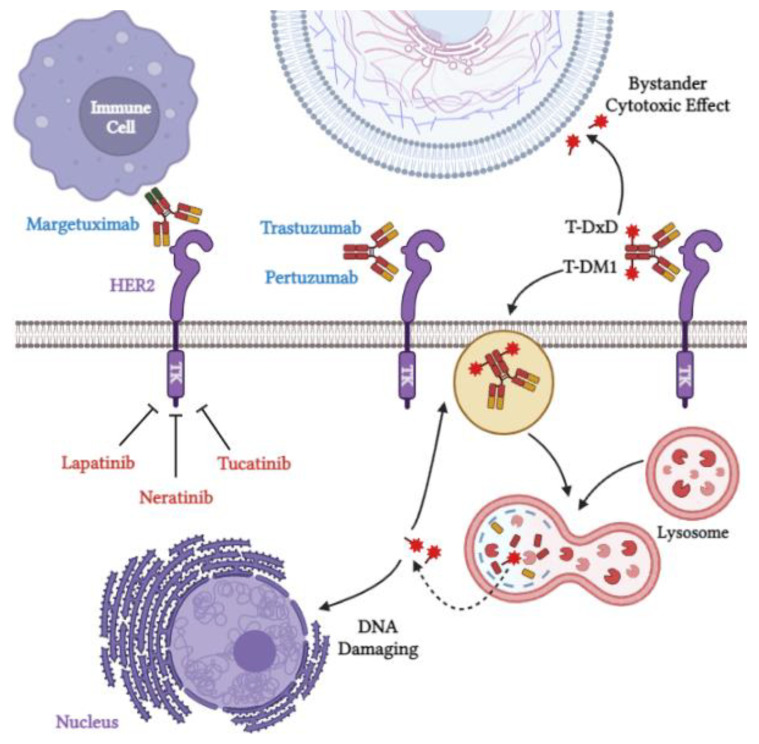
Mechanisms of action of some anti-HER2 therapies. Created with BioRender.com, accessed on 1 October 2022.

**Table 1 cancers-14-06173-t001:** Incidence of HER2 positivity in diverse cancers by IHC.

Tumor Type	HER2 Positivity (%), Reported by Yan et al. [30]	HER2 Positivity (%), Reported by Other Studies
Bladder cancers	12.4	16 [35]
Esophageal and esophagogastric junction cancers	11.3	14.9 [36]
Breast cancers	10.5	18.3 [37]
Gallbladder cancers	9.8	11.11 [38]
Cholangiocarcinomas (extrahepatic)	6.3	8.5 [39]
Gastric adenocarcinomas	4.7	17.3 [40]
Cervical cancers	3.9	1.5 [41]
Uterine cancers	3	6 [42]
Testicular cancers	2.4	5–8 [32]
Colorectal cancers	1.8	2 [43]
Ovarian (epithelial) cancers	1.6	8.16 [44]
Head and neck carcinomas	1.3	4–19 [45,46]
Lung cancers (non-small cells)	1.1	2.5 [47]
Intestinal (small) malignancies	0.9	3 [48]
Pancreatic adenocarcinomas	0.7	2 [49]
Cholangiocarcinomas (intrahepatic)	0.6	4 [50]
Prostate cancers	0.6	1.5 [51,52]
Hepatocellular carcinomas	0.4	0 [53]
Ovarian (non-epithelial) cancers	0.4	7.69 [54]
Melanomas	0.1	0 [55]
Gastrointestinal stromal tumors	0	0 [56]
Glioblastoma multiforme, high grade gliomas	0	0 [57]
Kidney cancers	0	2.3 [58]
Lung cancers (small-cells)	0	13 [59]
Melanomas (uveal)	0	Not found
Neuroendocrine tumors	0	0 [60]
Sarcomas (peritoneal, retroperitoneal)	0	Not found
Sarcomas (soft tissues)	0	8 [61]
Thymic cancers	0	0 [62]
Thyroid cancers	0	0 [63]
Gliomas (low-grade)	0	7 [57]
Oligodendrogliomas	0	Not found
Penile cancers	0	0 [64]
Pituitary cancers	0	5 [65]
Solitary fibrous tumors	0	Not found
Overall	2.7	Not found

**Table 2 cancers-14-06173-t002:** Clinical trials with CAR-T-cell therapy for HER2-positive tumors: (clinicaltrials.gov, accessed on 12 October 2022).

NCT Number	Study Title	Study Status	Conditions	Interventions	Phases
NCT03267173	Evaluate the Safety and Efficacy of CAR-T in the Treatment of Pancreatic Cancer.	UNKNOWN	Pancreatic Cancer	Drug: Mesothelin, PSCA, CEA, HER2, MUC1, EGFRvIII targeted and other CAR-T cell	EARLY_PHASE1
NCT02547961	Chimeric Antigen Receptor-Modified T Cells for Breast Cancer	WITHDRAWN (Project terminated due to revision of local regulations)	Breast Cancer	BIOLOGICAL: HER2-targeted CAR-T cells	PHASE1|PHASE2
NCT04511871	A Phase I Trial of CCT303-406 in Patients with Relapsed or Refractory HER2 Positive Solid Tumors	RECRUITING	Solid Tumor, Gastric Cancer, Breast Cancer, Ovarian Cancer, Sarcoma	BIOLOGICAL: HER2-targeted CAR-T cells	PHASE1
NCT04903080	HER2-specific Chimeric Antigen Receptor (CAR) T Cells for Children with Ependymoma	RECRUITING	Ependymoma	BIOLOGICAL: HER2-targeted CAR-T cells	PHASE1
NCT00902044	Her2 Chimeric Antigen Receptor Expressing T Cells in Advanced Sarcoma	ACTIVE_NOT_RECRUITING	Sarcoma	GENETIC: Autologous HER2-targeted CAR-T cells, DRUG: Fludarabine, DRUG: Cyclophosphamide	PHASE1
NCT02713984	A Clinical Research of CAR T Cells Targeting HER2 Positive Cancer	WITHDRAWN (Reform CAR structure due to safety consideration)	Breast Cancer, Ovarian Cancer, Lung Cancer, Gastric Cancer, Colorectal Cancer, Glioma, Pancreatic Cancer	BIOLOGICAL: HER2-targeted CAR-T cells	PHASE1, PHASE2
NCT03389230	Memory-Enriched T Cells in Treating Patients with Recurrent or Refractory Grade III-IV Glioma	RECRUITING	Glioblastoma, Malignant Glioma, Recurrent Glioma, Refractory Glioma, WHO Grade III Glioma	BIOLOGICAL: HER2-targeted CAR-T cells	PHASE1
NCT03696030	HER2-CAR T Cells in Treating Patients with Recurrent Brain or Leptomeningeal Metastases	RECRUITING	Malignant Neoplasm, Metastatic Malignant Neoplasm in the Brain, Metastatic Malignant Neoplasm in the Leptomeninges, Breast Cancer, HER2-positive Breast Cancer	BIOLOGICAL: HER2-targeted CAR-T cells	PHASE1
NCT04650451	Safety and Activity Study of HER2-Targeted Dual Switch CAR-T Cells (BPX-603) in Subjects with HER2-Positive Solid Tumors	RECRUITING	HER-2 Gene Amplification, HER2-positive Gastric Cancer, HER2-positive Breast Cancer, HER-2 Protein Overexpression, Solid Tumor, Adult	BIOLOGICAL: HER2-targeted CAR-T cells	PHASE1
NCT03740256	Binary Oncolytic Adenovirus in Combination with HER2-Specific Autologous CAR VST, Advanced HER2 Positive Solid Tumors	RECRUITING	Bladder Cancer, Head and Neck Squamous Cell Carcinoma, Cancer of the Salivary Gland, Lung Cancer, Breast Cancer, Gastric Cancer, Esophageal Cancer, Colorectal Cancer, Pancreatic Adenocarcinoma, Solid Tumor	BIOLOGICAL: HER2-targeted CAR-T cells	PHASE1
NCT04995003	HER2 Chimeric Antigen Receptor (CAR) T Cells in Combination with Checkpoint Blockade in Patients with Advanced Sarcoma	RECRUITING	Sarcoma, HER-2 Protein Overexpression	GENETIC: HER2-targeted CAR-T cells, DRUG: Pembrolizumab, DRUG: Nivolumab, DRUG: Lymphodepletion Chemotherapy	PHASE1
NCT04684459	Dual-targeting HER2 and PD-L1 CAR-T for Cancers with Pleural or Peritoneal Metastasis	RECRUITING	Peritoneal Carcinoma Metastatic, Pleural Effusion, Malignant	BIOLOGICAL: Dual-targeting HER2 and PD-L1 CAR-T cells	EARLY_PHASE1
NCT00889954	Her2 and TGFBeta Cytotoxic T Cells in Treatment of Her2 Positive Malignancy	COMPLETED	HER2 Positive Malignancies	BIOLOGICAL: TGFBeta resistant HER2/EBV-CTLs (CAR-T cells)	PHASE1
NCT00924287	Gene Therapy Using Anti-Her-2 Cells to Treat Metastatic Cancer	TERMINATED (This study was terminated after the first patient treated on study died as a result of the treatment.)	Metastatic Cancer	DRUG: HER2-targeted CAR-T cells plus IV IL-2, DRUG: Cyclophosphamide, DRUG: Fludarabine, DRUG: Mesna	PHASE1, PHASE2
NCT02442297	T Cells Expressing HER2-specific Chimeric Antigen Receptors (CAR) for Patients with HER2-Positive CNS Tumors	RECRUITING	Brain Tumor, Recurrent, Brain Tumor, Refractory	BIOLOGICAL: HER2-targeted CAR-T cells	PHASE1
NCT04430595	Multi-4SCAR-T Therapy Targeting Breast Cancer	RECRUITING	Breast Cancer	BIOLOGICAL: multiple 4th generation CAR-T cells targeted Her2, GD2, and CD44v6	PHASE1, PHASE2
NCT01109095	CMV-specific Cytotoxic T Lymphocytes Expressing CAR Targeting HER2 in Patients with GBM	COMPLETED	Glioblastoma Multiforme (GBM)	BIOLOGICAL: HER2-targeted CAR CMV-specific CTLs (CMV-specific cytotoxic T cells)	PHASE1
NCT03500991	HER2-specific CAR T Cell Locoregional Immunotherapy for HER2-positive Recurrent/Refractory Pediatric CNS Tumors	RECRUITING	Central Nervous System Tumor, Pediatric, Glioma, Ependymoma, Medulloblastoma, Germ Cell Tumor, Atypical Teratoid/Rhabdoid Tumor, Primitive Neuroectodermal Tumor, Choroid Plexus Carcinoma, Pineoblastoma	BIOLOGICAL: HER2-targeted CAR-T cells	PHASE1
NCT03198052	PSCA/MUC1/TGFOI/HER2/Mesothelin/Lewis-Y/GPC3/AXL/EGFR/B7-H3/Claudin18.2-CAR-T Cells Immunotherapy Against Cancers	RECRUITING	Lung Cancer, Cancer, Immunotherapy, CAR-T Cell	BIOLOGICAL: CAR-T cells targeting PSCA, MUC1, TGFOI, HER2, Mesothelin, Lewis-Y, GPC3, AXL, EGFR, Claudin18.2, or B7-H3	PHASE1

**Table 3 cancers-14-06173-t003:** Mechanisms of resistance to HER2-targeted therapies.

Resistance-Causing Factor	Mechanism of Resistance	Strategies to Overcome Resistance	Resource
HER2(L755S) mutation	Activating mutation	Second-generation TKIs (neratinib)	[144,148]
Overexpression of HER1 and HER3	Enhancing the affinity of HER2/HER3 and HER1/HER3 and reducing HER2 binding to neratinib	Combined anti-HER2, PI3K inhibition and TKIs (neratinib and lapatinib)	[144,146,148,149]
Generation of p95^HER2^	A truncated form of HER2 that lacks the ECD but retains kinase activity	Combined chemotherapy (paclitaxel), trastuzumab and TKIs (lapatinib)	[144,148]
Overexpression of mucin 4 (MUC4) and the CD44–hyaluronan polymer complex	Masking the HER2 epitope as well as stabilizing and activating HER2	Combined soluble TNF inhibitors, trastuzumab and TKIs (lapatinib)	[144,146,148,149]
Loss of PTEN	Hyperactivation of the mTOR pathway	Combined PI3K trastuzumab and pertuzumab Combination of PI3K and MEK inhibitors	[144,145,146,148,149]
PIK3CA mutations	Increased and unregulated activation of the PI3K pathway	T-DM1 PI3K inhibitors in combination with trastuzumab and pertuzumab Combined mTOR inhibitor (everolimus), trastuzumab and chemotherapy	[144,145,146,148,149]
Expression of estrogen receptors	Offering an escape from HER2 signaling inhibition insuring tumor survival	Concomitant inhibition of both ER and HER2 signaling	[144,148]
Overexpression of Cyclin D1 and/or CDK4/6	Activating cell proliferation	Combination blocking of HER2 and ER and CDK4/6–cyclin D1 activation	[148,149]
RAS–MAPK activating mutations	Sustained activation of RAS–MAPK signaling	MEK–ERK inhibitors	[148]
Heterogeneous expression of HER2	Subclone lacking the target could escape the effects of the targeted therapy and lead to tumor recurrence	Potent HER2-targeted agents (T-DXd)	[144,148,150]
Expressing high levels of HLA g by tumor cells	Inhibiting NK cells through the engagement of killer cell immunoglobulin-like receptors (KIRs)	Combined blockade of HLA g and PDL1/PD1	[148]
Overexpression of CD47 by tumor cells	Inhibiting phagocytosis	Combination of magrolimab (mAb that targets CD47) and trastuzumab	[148]
Expression of CDK12- or RAC1 by tumor cells	Activating cell proliferation	Combinations of HER2 TKIs, CDK12 and RAC1 inhibitors	[148]
Increased activity and expression of the drug efflux pump	Reducing the cytotoxic effect of T-DM1	Combination of T-DM1 and pump inhibitors	[144]
c-MET hyperactivity or amplification	Inducing HER3-mediated activation of PI3K Sustained Akt activation	c-MET inhibitors	[145,146,149]
Overexpression of IGF1R	Activation of HER2 Inducing degradation of p27	IGF1R signaling inhibition	[145,146]
Src activation	Inhibiting PTEN	Combination of trastuzumab and Src inhibitor (dasatinib)	[145,149]
Suppression of the PP2A family	Sustained activation of the PI3K/AKT/mTOR pathway	Combination of EZH2 inhibitor with HER2-targeted therapy	[145]
Upregulation of miR-221	Inhibiting PTEN Targeting p57 and p27	Src inhibitors	[145]
AXL overexpression	Activation of PI3K/AKT and MAPK pathways in a ligand-independent manner	AXL inhibitor plus trastuzumab	[151]
Upregulation of MCL-1	Inhibition of apoptosis	PARP inhibitor (olaparib)	[149,152]
Co-expression of HER2(T862A) and HER2(L755S) mutations	Enhancing HER2 activation and impairing TKIs sensitivity	Combined inhibition of HER2 and MEK	[153]
Activating mutations in TGF-β	Enhancing HER ligand shedding	Trastuzumab, pertuzumab and TGFꞵ inhibitors	[146,149]

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
