# Peer review of "The Role of Tumor-Associated Antigen HER2/neu in Tumor Development and the Different Approaches for Using It in Treatment: Many Choices and Future Directions"

_cancers, 2022, doi:10.3390/cancers14246173_

Round 1

Reviewer 1 Report

Manuscript ID: cancers-1975358

Type: Review

Title: The role of tumor-associated antigen HER2/neu in tumor development and the different approaches for using it in treatment: many choices and future directions

General Comments: Authors presented the current literature on Her2 in various cancers and the clinical trials with agents against Her2-expressing cancers. There are many reviews on Her2 role in cancers and the current and future management of Her2 cancers. Though the authors tried to assemble the clinical trials and results, this review needs refinement. Please see below for specific comments.

  1. The title and other information provided in the body suggest that this review was written in the broader sense, where the role of Her2 was discussed in various cancers. Whereas, specifically a paragraph was written for breast cancer. Include a brief description of different cancers and Her2 expression/role individually.  
  2. Provide a table and a paragraph in the body of the review on the expression of Her2 in various cancers at different stages [precancer lesions/adenomas/adenocarcinomas/metastases] of cancer [ all cancers].
  3. There are many reviews on Her2 in breast and other cancers. The reader expects new important information on the mechanisms of Her2 resistance and how to overcome Her2 resistance observed in various cancers based on preclinical and clinical trials. Please provide your thoughts and include this information in the text.
  4. List the other possible therapies to overcome Her2 resistance and increase response rate in patients without toxicity. List lessons learned from the clinical trials. Include this information in a separate paragraph.  
  5. In Table 1- please include the type of patients recruited. And whether the treatment worked in all patients, if it did not work in a few patients, explain why? Explain these details in the body of the review.

Author Response

We would like to thank the Reviewer for taking the necessary time and effort to revise the manuscript. We sincerely appreciate all the valuable comments and suggestions, which helped us in improving the quality of the work.

All of the recommendations have been addressed in the manuscript with the exception of number 5, as we looked up all of these trials and no published results were found.

Reviewer 2 Report

This is a comprehensive review for current therapy and promising future directions for Her2-oriented therapy. There are many Her2-related reviews published before. This review has an emphasis on using Her2 as a tumor associated antigen for immunotherapy and included the most current development knowledge in this field. The review is well written and I did not have any further suggestions.

Author Response

We would like to thank the Reviewer for taking the necessary time and effort to revise the manuscript. We sincerely appreciate all the valuable comments on our manuscript.

Reviewer 3 Report

The grammar/style must be improved.  Please consider joining forces with a native or advanced english speaker.  There are too many corrections to mention.

For example:

Abstract: "considered a poor prognosis" is better "considered a predictor of poor prognosis"  and "starting with t..." instead of "started with..."

Line 23 "Not to mention" Is not a smooth way to start a sentence. Could use. "Particularly relevant is...."

On content, I believe is better to rephrase "deficient signaling" since in its natural form HER2 is not deficient.  Be specific. Do you mean hyperactive or which signaling pathways are deregulated?, in which type of cancer? provide citations.

Author Response

We would like to thank the Reviewer for taking the necessary time and effort to revise the manuscript. We sincerely appreciate all the valuable comments and suggestions, which helped us in improving the quality of the work.

The manuscript has been edited, hoping the grammar and the style now match standards of the journal.

Reviewer 4 Report

This is an interesting review article which comprehensively addresses the functional role of HER-2 in oncogenic pathways and its role as therapeutic target in breast cancer including anti HER-2 monoclonal antibodies, HER-2 antibody-drug conjugates, HER-2 targeting adoptive T-cell therapies, HER-2 TKIs and HER-2-based vaccines. The quality of this review article could be improved providing the authors would address the following points:

1.The tumor microenvironment in breast cancer and other types of cancer provides a serious obstacle for the clinical efficacy of (immune)therapeutic treatments. To this end, the interplay between cancer cells and the tumor microenvironment establishes mechanisms inhibiting or promoting an endogenous or drug-induced antitumor immune response. Therefore, it will be useful, if the authors could briefly address this issue also in the context of HER-2 expression.  

2. Because of the situation described in the paragraph above, combination regimens are being employed to offer novel modalities for altering the tumor microenvironment in a way to induce tumor cell killing, for instance by combining immune checkpoint inhibitors with other therapeutic modalities such as or HER-2 antibodies or chemotherapeutic regimens, vaccines etc. The authors should address this important issue.

3.It will be also useful to discuss the role of PD-L1 expression in HER2+ breast cancer molecular subtypes for defining patients likely to respond to immune checkpoint inhibition-based immunotherapies.

3. The role of HER2 antibodies or HER2 – drug conjugates, such as trastuzumab deruxtecan, for the treatment of HER-2 -  low tumors should be also discussed.

Author Response

We would like to thank the Reviewer for taking the necessary time and effort to revise the manuscript. We sincerely appreciate all the valuable comments and suggestions, which helped us in improving the quality of the work.

All of the recommendations have been addressed in the manuscript.

Round 2

Reviewer 1 Report

Authors addressed 4 out of 5 the queries. Please review the literature thoroughly for the available articles [only few cancer types of data may be available], see below publications which have listed the Her 2 clinical trail data. Kindly search and review the individual articles and complete the table 1. [In Table 1- please include the type of patients recruited. And whether the treatment worked in all patients, if it did not work in a few patients, explain why? Explain these details in the body of the review].

 Publications: Examples

1.       Neoadjuvant dual HER2-targeted therapy with lapatinib and trastuzumab improves pathologic complete response in patients with early stage HER2-positive breast cancer: a meta-analysis of randomized prospective clinical trials. PMID: 25732265

2.       Zhou C, Li X, Wang Q. Pyrotinib in HER2-mutant advanced lung adenocarcinoma after platinum-based chemotherapy: a multicenter, open-label, single-arm, phase II study. J Clin Oncol. 2020;38(24):2753-61.

3.       Li BT, Shen R, Buonocore D, Olah ZT, Ni A, Ginsberg MS, et al. Ado-Trastuzumab Emtansine for patients with HER2-mutant lung cancers: results from a phase II basket trial. J Clin Oncol. 2018;36:2532-7.

4.       Dual neoadjuvant blockade plus chemotherapy versus monotherapy for the treatment of women with non-metastatic HER2-positive breast cancer: a systematic review and meta-analysis. Clin Transl Oncol. 2022 Nov 22 PMID: 36417083

Author Response

      We would like to thank the reviewer for the helpful suggestion, but there may be some confusion because table 2 in the manuscript summarizes clinical trials that use CAR-T cell therapy in HER2-positive cancers, whereas the publications provided by the reviewer discuss clinical trials that use other types of HER2-targeted therapies such as monoclonal antibodies, tyrosine kinase inhibitors, and antibody-drug conjugates. It is also unclear what the reviewer means by “the type of patients recruited”.

     We responded to the reviewer's recommendation in the first round of revision by explaining that we sought for the results of the mentioned clinical trials but found none. We double-checked the Clinical trials.gov and Pubmed websites for results, but nothing was found. We have provided the reasons of termination or withdrawal of couple of studies.